# Possible Three-Dimensional Topological Insulator in Pyrochlore Oxides

**Izumi Hase ***  **and Takashi Yanagisawa**

National Institute of Advanced Industrial Science and Technology (AIST), Tsukuba Central 2, Umezono 1-1-4, Tsukuba, Ibaraki 305-8568, Japan; t-yanagisawa@aist.go.jp
* Correspondence: i.hase@aist.go.jp; Tel.: +81-298-61-5147

**Abstract:** A Kene–Mele-type nearest-neighbor tight-binding model on a pyrochlore lattice is known to be a topological insulator in some parameter region. It is an important task to realize a topological insulator in a real compound, especially in an oxide that is stable in air. In this paper we systematically performed band structure calculations for six pyrochlore oxides $A_2B_2O_7$ (A = Sn, Pb, Tl; B = Nb, Ta), which are properly described by this model, and found that heavily hole-doped $Sn_2Nb_2O_7$ is a good candidate. Surprisingly, an effective spin–orbit coupling constant $\lambda$ changes its sign depending on the composition of the material. Furthermore, we calculated the band structure of three virtual pyrochlore oxides, namely $In_2Nb_2O_7$, $In_2Ta_2O_7$ and $Sn_2Zr_2O_7$. We found that $Sn_2Zr_2O_7$ has a band gap at the $k = 0$ ($\Gamma$) point, similar to $Sn_2Nb_2O_7$, though the band structure of $Sn_2Zr_2O_7$ itself differs from the ideal nearest-neighbor tight-binding model. We propose that the co-doped system $(In,Sn)_2(Nb,Zr)_2O_7$ may become a candidate of the three-dimensional strong topological insulator.

**Keywords:** pyrochlore oxide; band calculation; flat band; spin–orbit interaction; topological insulator

## 1. Introduction

Topologically non-trivial states in materials have been paid much attention in the last decades. Among them, topological insulators (TIs) have attracted attention, not only from a basic but also from an application point of view. In TI, while the bulk is insulating, the surface has a metallic state. Since this surface state is robust against perturbations that do not violate the time reversal symmetry, it has been pointed out that it can be applied to a quantum computing device. Considering such applications, the material is desired to be three-dimensional and stable in the air, namely oxide. However, very little is known about oxides that are candidates for TI.

Outstanding works by Kane and Mele [1,2] have revealed that graphene may show a quantum spin Hall (QSH) effect if the spin–orbit coupling (SOC) is sufficiently strong. This effect can be explained as follows: Graphene has Dirac points that are protected by space–group symmetry, and this degeneracy is lifted by the SOC. This SOC opens a gap and makes the system a TI. A conventional (quantum) Hall effect is caused by an external magnetic field, while in this case the SOC plays the role of "magnetic field", depending on the direction of the electron spin. Although the SOC in real graphene is too small to open a gap, subsequent works predicted that the HgTe–CdTe quantum well is a good candidate showing a QSH effect [3]. This prediction has been subsequently verified experimentally [4].

Later on, the Kane–Mele model [1,2] has been extended to other lattice models. Guo and Franz applied this model to a two-dimensional (2D) Kagome lattice [5] and three-dimensional (3D) pyrochlore lattice [6]. Both of these models are known to have a flat band (FB) that has no energy dispersion when only the nearest neighbor transfer is considered. Adding SOC with the type of Kane–Mele model, these lattice models show various topological properties.

The Guo and Franz (GF) model can be described as follows [6]:

$$H = -t \sum_{\langle ij \rangle \sigma} c_{i\sigma}^{\dagger} c_{j\sigma} + i\lambda \sum_{\ll ij \gg \alpha\beta} \left( \boldsymbol{d}_{ij}^{1} \times \boldsymbol{d}_{ij}^{2} \right) \cdot \boldsymbol{\sigma}_{\alpha\beta} c_{i\alpha}^{\dagger} c_{j\beta} \tag{1}$$

where the first term of Equation (1) denotes the non-interacting tight-binding Hamiltonian, and *<ij>* denotes the nearest neighbors. This lattice sum is taken on the pyrochlore lattice shown in Figure 1a. This term can also be regarded as the Mielke model without an electron–electron interaction [7,8]. The second term denotes SOC with an effective coupling constant λ. $\boldsymbol{\sigma}_{\alpha\beta}$ denotes the spin αβ component of the Pauli matrices. In the pyrochlore lattice there are four sites in the primitive cell, and we obtain four bands. Here, $\boldsymbol{d}_{ij}^{1,2}$ denote the nearest-neighbor vectors traversed between the second neighbor sites *i* and *j*. The summation *<<ij>>* is taken for these second neighbor sites. In case of λ = 0, this Hamiltonian is rigorously diagonalized and the eigenvalues are

$$E_k^{1,2} = -2t \left[ 1 \pm \sqrt{1 + A_k} \right], \ E_k^{3,4} = 2t \tag{2}$$

where $A_k = \cos(2k_x)\cos(2k_y) + \cos(2k_y)\cos(2k_z) + \cos(2k_z)\cos(2k_x)$. $E^3$ and $E^4$ do not depend on the wave vector $\boldsymbol{k}$, so they are called flat bands (FBs). These energy eigenvalues are shown in Figure 1b. Note that these FBs touch the dispersive band $E_k^2$ at $\boldsymbol{k} = 0$. In other words, the state at $\boldsymbol{k} = 0$ is three-fold (six-fold if we consider spin) degenerated. Therefore, in case of half-filling, i.e., two of the four bands are filled, the system is a zero-gap semiconductor. FB itself induces many attractive physical properties [7–10], but in this paper we focus on the topological aspect of this model.

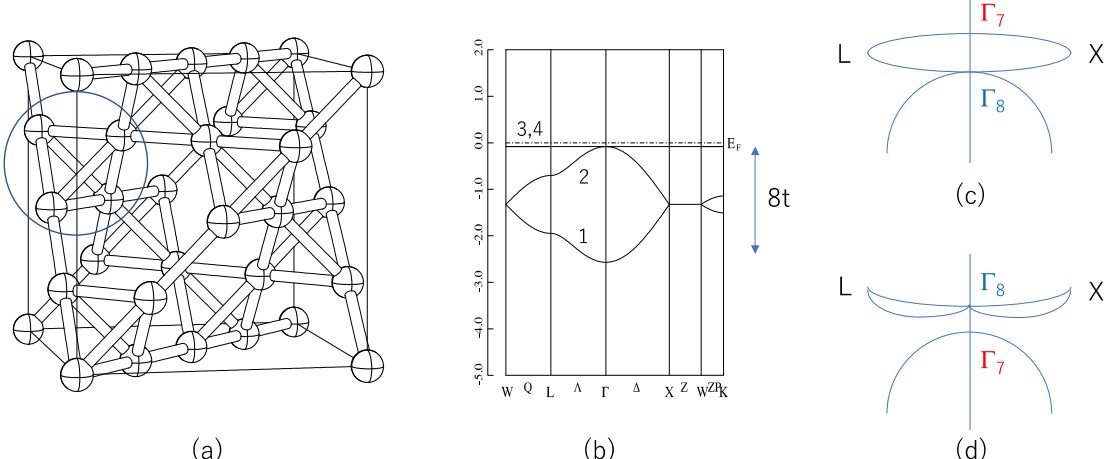

(a)                    (b)                    (c)                    (d)

**Figure 1.** (**a**) Pylochlore lattice in the face-centered cubic Bravais lattice. The large circle shows the 4 atomic sites in the primitive unit cell, forming a regular tetrahedron. These tetrahedra shear corners and form a three-dimensional network; (**b**) energy dispersion of the Guo and Franz (GF) model with λ = 0 (Equation (2)). Numbers 1–4 denote the band index. Energy offset is added to simulate the band structure of $Pb_2Ta_2O_7$. Note that the total bandwidth is large (= 8t), while the flat bands (FBs) ($E^3$ and $E^4$) does not have any dispersion; (**c**) a schematic picture of the energy dispersion of the GF model for λ > 0. At the Γ point the degeneracy is partly lifted, but at the X and L points the degeneracy is protected by symmetry, so the system remains gapless; (**d**) a schematic picture of the energy dispersion of the GF model for λ < 0. In this case the system opens a gap, and this becomes a strong TI [6].

Next, we include SOC. This SOC lifts the abovementioned six-fold degeneracy into two-fold ($\Gamma_7$) and four-fold ($\Gamma_8$) degenerated states. Which of these states have higher energy depends on the sign of λ. A schematic picture is shown in Figure 1c,d. When λ > 0, the energy of $\Gamma_7$ is higher than that of $\Gamma_8$, i.e., $E(\Gamma_7) > E(\Gamma_8)$. In this case, four-fold degenerated FBs split into $\Gamma_7$ and $\Gamma_8$ at the Γ point, but are still degenerated at the zone boundary points X and L. Therefore, the system remains gapless when

$\lambda > 0$. On the contrary when $\lambda < 0$, the energy of $\Gamma_7$ is lower than that of $\Gamma_8$, i.e., $E(\Gamma_7) < E(\Gamma_8)$. In case of half-filling and $\lambda < 0$, the system has a band gap $\Delta = 24|\lambda|$. Guo and Franz [6] have shown that this is a 3D strong TI having topological indices (1;000), where $(\nu_0;\nu_1\nu_2\nu_3)$ denote the four-component $Z_2$ topological indices [11,12]. Each $\nu_i$ has a binary value 0 or 1, and if all $\nu_i$ are zero, i.e., (0;000), the system is an ordinary band insulator; if $\nu_0 = 1$ it is a strong TI, which means topologically protected surface states exist on all surfaces and they are robust with regard to non-magnetic disorder.

It is very important to apply these results to real materials. In this sense, the GF model defined on the pyrochlore lattice is very attractive since the pyrochlore oxides $A_2B_2O_7$ include two independent pyrochlore sublattices [13]. However, even though there are hundreds of pyrochlore oxides [13], there are very few compounds whose electronic states near the Fermi level ($E_F$) can be approximated by the GF model. The reason for this is as follows: In the GF model, the transfer integral $t$ is isotropic, while in most of the pyrochlore oxides the relevant orbital is an anisotropic d- or f- orbital.

In previous papers [14–16] we have shown that six pyrochlore oxides $A_2B_2O_7$ (A = Sn, Pb, Tl; B = Nb, Ta) have band structures that are well described in the GF model. In those papers, we mainly focused on the possible ferromagnetism induced by the quasi-FB. We have already shown [14] that $Sn_2Nb_2O_7$ has $E(\Gamma_7) < E(\Gamma_8)$, i.e., $\lambda < 0$. On the other hand, $Sn_2Ta_2O_7$ has $\lambda > 0$. It means that $Sn_2Nb_2O_7$ is a good candidate for a strong TI when the holes are sufficiently doped ($Sn_2Nb_2O_7$ itself is away from half-filling), but $Sn_2Ta_2O_7$ is not. After this work, Zhou et al. proposed a novel topological state for ferromagnetic $Sn_2Nb_2O_7$ induced by quasi-FB [17], and Zhang et al. proposed that $Tl_2Nb_2O_{6+x}$ ($0 \le x \le 1$) leads to various topological phases accompanied with lattice distortion [18].

The purpose of this paper is to extend the above discussion to other pyrochlore oxides of which the band structure near $E_F$ is approximated by the GF model. We calculated the band structure of six pyrochlore oxides $A_2B_2O_7$ (A = Sn, Pb, Tl; B = Nb, Ta) from first principles. Additionally, we calculated the band structure of three hypothetical compounds, $In_2Nb_2O_7$, $In_2Ta_2O_7$ and $Sn_2Zr_2O_7$, which may be potential candidates for a TI. We found that only $Sn_2Nb_2O_7$ and $Sn_2Zr_2O_7$ gives $\lambda < 0$, though the band structure of $Sn_2Zr_2O_7$ cannot fit very well with the GF model.

This paper is organized as follows. Section 2 explains the method of calculation. The main results and discussions are shown in Section 3. Conclusions are given in Section 4.

## 2. Materials and Computational Methods

We calculated six pyrochlore oxides with the composition $A_2B_2O_7$ (A = Sn, Pb, Tl; B = Nb, Ta) from first principles. We used the density-functional theory (DFT) and a linearized augmented plane wave with the addition of a local orbital (LAPW+lo) scheme (WIEN2k code [19]). The exchange-correlation potential was constructed within the general gradient approximation [20]. The $k$-point mesh was set so that the total number of the mesh was about 1000 in the first Brillouin zone. The parameter $RK_{max}$, which is a product of the maximum radius of the muffin-tin spheres $R$ and the cut-off wave number of the plane-wave basis $K_{max}$, was set to be 7.0. For simplicity, we assumed that the calculated compounds all have an ideal $A_2B_2O_6O'$ pyrochlore structure with the space group Fd-3m (#227). Since oxygen atoms occupy two crystallographic sites, we named these sites as O and O' to distinguish between them. The atomic positions are A(0, 0, 0), B(1/2, 1/2, 1/2), O($u$, 1/8, 1/8) and O'(1/8, 1/8, 1/8). This pyrochlore structure contains 88 atoms in the conventional cubic unit cell. In this work we focus on the A atoms, which form the pyrochlore lattice shown in Figure 1a. O' atoms are at the center of the $A_4$ tetrahedron shown by a circle in Figure 1a. We optimized this parameter $u$ by minimizing the Hellmann–Feynman force $\boldsymbol{F} = -\left\langle \psi_{\boldsymbol{R}} \middle| \frac{\partial H}{\partial \boldsymbol{R}} \middle| \psi_{\boldsymbol{R}} \right\rangle$, where $H$ denotes the one-electron Hamiltonian for the atomic coordinates $\boldsymbol{R}$, and $\psi_{\boldsymbol{R}}$ is its eigenfunction. The convergence of the atomic positions was judged by this Hellmann–Feynman force working on each atom, which was to be less than 1.0 mRy/a.u. We also optimized the lattice constant $a$ for $Tl_2Ta_2O_7$. The optimized value $a = 10.716$ Å well agreed with an experimental value $a = 10.56$ Å [21] or 10.651 Å [22]. For the other five compounds, we used the experimental lattice constant.

After the relaxation of the atomic position and lattice constant, a spin–orbit interaction (SOI) is included via a second-variational step using a scalar-relativistic eigenfunction as the basis [23,24]. Hereafter, we distinguish between SOI and SOC. The former is essentially an on-site term, explicitly included in the band calculation [19,23,24]. The latter is the term introduced in the Kane–Mele model [1] and GF model [6], which is essentially a non-local term. We can estimate the SOC from the band calculation by fitting the bands to the GF model, or more simply, just see the energy splitting at the $\Gamma$ point $\Delta_{\text{SO}} = E(\Gamma_7) - E(\Gamma_8)$ and use the relation $\Delta_{\text{SO}} = 24\,\lambda$ [6].

In order to understand the electronic structure of this $A_2B_2O_7$ system, we also performed band calculations on the virtual compounds $In_2Nb_2O_7$, $In_2Ta_2O_7$ and $Sn_2Zr_2O_7$. For these compounds, we relaxed both of the lattice constant and the atomic positions. The obtained lattice constant is $a = 10.611$, 10.631 and 10.618 Å for $In_2Nb_2O_7$, $In_2Ta_2O_7$ and $Sn_2Zr_2O_7$, respectively. They are all within the normal range of the lattice constants of pyrochlore oxides [13].

## 3. Results and Discussions

First, we show the band structure of $Pb_2Ta_2O_7$ in Figure 2a. Similar to other quasi-FB oxides [14–18], this compound also shows a characteristic quasi-FB just below $E_F$. The origin of this quasi-FB is explained as follows. The formal chemical valences of $Pb_2Ta_2O_7$ are denoted as $Pb^{2+}_2Ta^{5+}_2O^{2-}_7$. Since $Ta^{5+}$ and $O^{2-}$ form closed shells, we see that only $Pb^{2+}$ is chemically active in the first approximation. The electron configuration of $Pb^{2+}$ is $(6s)^2$, so we can expect that the band gap is opened between the Ta5d band and Pb6s band. This is verified by density-of-states (DOS) analysis shown in Figure 2b. We see that the sharp peak coming from quasi-FB mainly consists of the Pb-s and O'-p components.

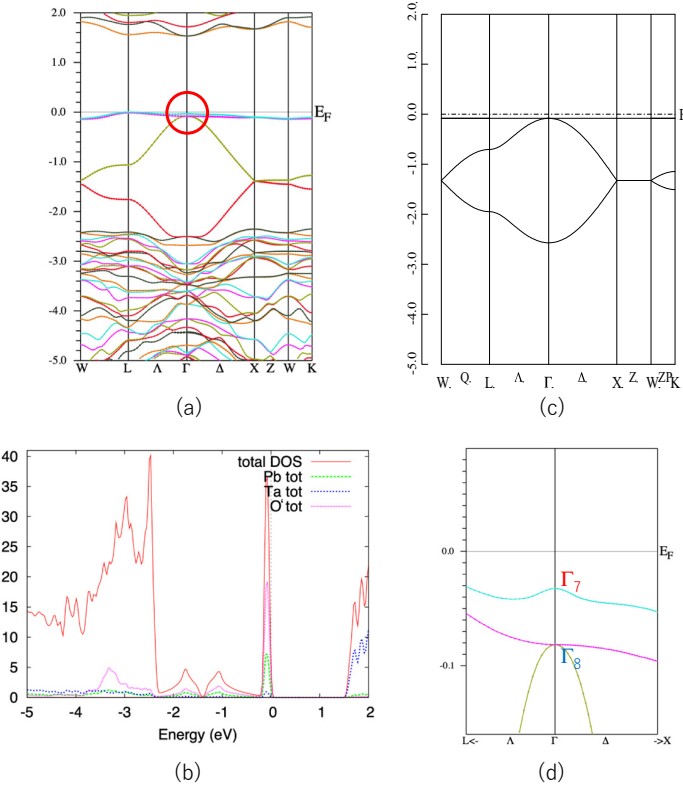

(a) (c)

(b) (d)

**Figure 2.** (**a**) Band structure of $Pb_2Ta_2O_7$. (**b**) DOS curve of $Pb_2Ta_2O_7$. (**c**) Energy dispersion of the GF model with $\lambda = 0$. (**d**) Blow-up figure of the valence band of $Pb_2Ta_2O_7$ near $\Gamma$ point, shown in the circle on Figure 1a. The units of the vertical axes are eV for (**a**), (**c**) and (**d**), and eV$^{-1}$ for (**b**).

As is explained in Section 1, a simple nearest-neighbor tight-binding model on a pyrochlore lattice gives a flat band as in Equation (2). We show the energy dispersion of this model in Figure 2c.

The agreement between the actual valence band of $Pb_2Ta_2O_7$ and the simple GF model is quite good, considering that the GF model includes only one parameter, $t$ (in Figure 2c we set $\lambda = 0$). Since the s-orbital is isotropic, the hopping integral $t$ in Equation (1) is also isotropic. Note that in most of the pyrochlore oxides the relevant (frontier) orbitals are d- or f-orbitals, which are anisotropic. We can also show that the more precise 10-orbital model, including four Pb-s and 6 O'-p orbitals, also have a FB [14]. However, we deal with the minimal 4-orbital model because we can utilize the results for the GF model.

Note that the top of the valence band of $Pb_2Ta_2O_7$ is not a complete FB but a quasi-FB. This small dispersion (about 0.2 eV) is attributed to the hopping terms other than the nearest neighbor one, such as next-nearest-neighbor hopping. However, this small dispersion does not change the topology of the band structure and we can use the result for the GF model.

We can see that the quasi-FB in $Pb_2Ta_2O_7$ is slightly split due to the SOI. Other bands, e.g., the bands at energy ~1.5 eV, split more than the quasi-FB at the valence band. This is because the conduction band mostly consist of Ta5d orbitals in which the SOI is large. If the valence band purely consists of Pb6s orbitals, the SOI should be zero. Small splitting of the quasi-FB in $Pb_2Ta_2O_7$ is due to the hybridization of the other orbitals. In Figure 2d we show a blow-up of the band structure of the $Pb_2Ta_2O_7$ near $E_F$ (shown by a circle in Figure 2a).

As shown in Reference [6], the GF model in the half-filled case has a band gap at the $\Gamma$ point when the SOC is switched on and $\lambda < 0$; in this case the system becomes a strong TI with $Z_2$ indices (1;000). On the other hand, it does not have a band gap when $\lambda > 0$. We see that $E(\Gamma_7)$ is larger than $E(\Gamma_8)$, i.e., $\lambda > 0$ in $Pb_2Ta_2O_7$. This situation corresponds to the case in Figure 1c. Note that these $Z_2$ indices are purely topological ones and robust for the small perturbations. Therefore, we can expect that these indices are unchanged in the real compounds we calculated, unless there are extra band crossings. Above these reasons, we can investigate the topological properties of the $A_2B_2O_7$ compounds by calculating the energy difference $\Delta_{SO} = E(\Gamma_7) - E(\Gamma_8)$ at the $\Gamma$ point ($k = 0$).

Figure 3 shows the energy difference $\Delta_{SO} = E(\Gamma_7) - E(\Gamma_8)$ in eight pyrochlore oxides $A_2B_2O_7$ (A = In, Sn, Tl, Pb; B = Nb, Ta), including two virtual oxides, $In_2Nb_2O_7$ and $In_2Ta_2O_7$. Apparently, only $Sn_2Nb_2O_7$ has a band gap, and the other seven oxides do not have a band gap. This result shows that $Sn_2Nb_2O_7$ may become a strong TI when the chemical potential is put in this band gap by heavily doping the holes. At present, hole doping in $Sn_2Nb_2O_7$ has been successful [25], but such a heavy doping is difficult. As for $Sn_2Nb_2O_7$, combining the topological band structure and the possible ferromagnetism due to quasi-FB, the existence of a novel three-dimensional Weyl point is suggested [17].

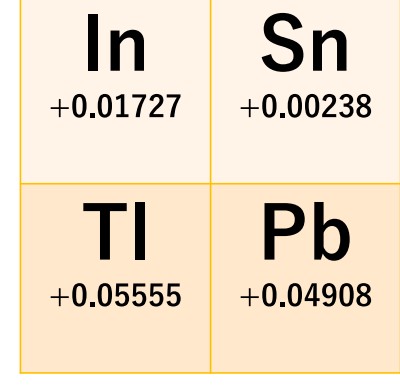

| In | Sn | | In | Sn |
|:---:|:---:|---|:---:|:---:|
| +0.00653 | -0.00503 | | +0.01727 | +0.00238 |
| **Tl** | **Pb** | | **Tl** | **Pb** |
| +0.05252 | +0.04939 | | +0.05555 | +0.04908 |
| B = Nb | | | B = Ta | |

**Figure 3.** Effective spin–orbit coupling constant $\lambda$ for eight $A_2B_2O_7$ pyrochlore oxides (units are eV). The left panels are for B = Nb, and the right panels are for B = Ta. A-site elements are arranged in the order of the periodic table. For example, the upper-left column in the left panel means $In_2Nb_2O_7$.

As seen in Figure 3, $\Delta_{SO}$ (or effective SOC coefficient $\lambda = \Delta_{SO}/24$) is material-dependent, even in its sign. We can see some trends in this figure:

1. When A is the period-6 element of the periodic table, $\Delta_{SO}$ is large, and when it is a period-5 element, $\Delta_{SO}$ is small. The difference of $\Delta_{SO}$ is about 40–50 meV.

2. When A is the group-13 element of the periodic table, $\Delta_{SO}$ is large, and when it is a period-14 element, $\Delta_{SO}$ is small. The difference of $\Delta_{SO}$ is about 3–15 meV.

3. $\Delta_{SO}$ does not so much depend on the B element. The difference of $\Delta_{SO}$ is about $-0.3$–10 meV.

Among these three trends, the first one is most crucial. It suggests that element A should be period-5 like In or Sn for obtaining a small $\Delta_{SO}$. In order to realize a TI in pyrochlore oxide, we should also tune the chemical potential. Half-filling of the 4-orbital GF model means that two of the four bands are occupied, i.e., there are four electrons in the primitive unit cell. Since there are four atomic sites, it turns so that the outermost electron configuration in the real compound is $(5s)^1$ or $(6s)^1$, which means there is one electron in the outermost s-orbital. We call this type of compound the "s1 compound". This includes $Tl_2Nb_2O_7$ and so on. A famous mother compound of the high-$T_c$ superconductor $BaBiO_3$ is also an s1 compound [26–29], but $Tl_2Nb_2O_7$ has a quasi-FB due to the strong geometric frustration of the pyrochlore lattice. This strong frustration may prevent forming a charge-density wave, which is seen in $BaBiO_3$. On the contrary, a compound containing two electrons in the outermost s-orbital is called the "s2 compound". As for realizing the TI, the s1 compound is desirable in order to realize the TI, though in general the s1 compound is not so stable.

From the above discussion, we consider that a virtual compound $Sn_2Zr_2O_7$ may be a good candidate for a TI because it is formally described as $Sn^{3+}_2Zr^{4+}_2O^{2-}_7$, having an s1 ion $Sn^{3+}$ and the other ions form a closed shell. This compound may also have small $\lambda$ since it contains an Sn atom on the A site. The calculated band structure of the $Sn_2Zr_2O_7$ is shown in Figure 4a,b. As is expected, we found $\Delta = 2.5$ meV, which means there is a band gap at the $\Gamma$ point. However, the valence band $Sn_2Zr_2O_7$ is apparently not described well by the GF model, as shown in Figure 4a. The reason why $Sn_2Zr_2O_7$ has such a band structure is not clear, but we note that the band structure of the other pyrochlore oxides with a formal ionic valence $A^{3+}_2B^{4+}_2O^{2-}_7$ (e.g., $Bi_2Ti_2O_7$) are not described well by the GF model (or non-interacting Mielke model) [15].

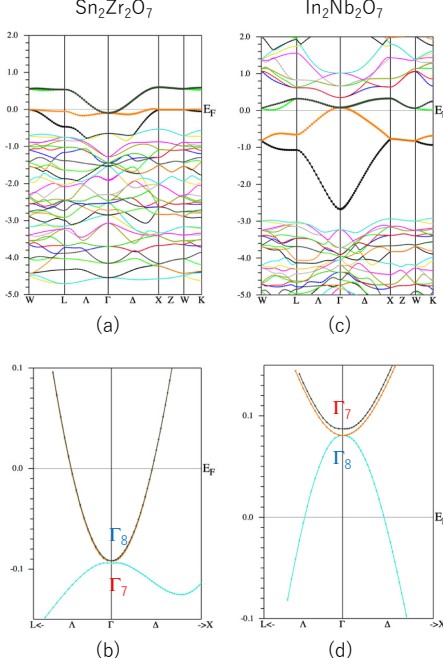

**Figure 4.** (**a**) Band structure of $Sn_2Zr_2O_7$ and (**b**) its blow-up figure of the valence band near the $\Gamma$ point. (**c**) Band structure of $In_2Nb_2O_7$ and (**d**) its blow-up figure of the valence band near the $\Gamma$ point. The units of the vertical axes are eV.

On the other hand, another virtual compound, $In_2Nb_2O_7$, has a band structure like the GF model (see Figure 4c), but $\lambda > 0$ (though not too large, see Figure 3). $Sn_2Zr_2O_7$ has $\lambda < 0$ but the band structure is not like the GF model. Therefore, we propose that a co-doped system, $(In,Sn)_2(Nb,Zr)_2O_7$, may have both of the desired properties for a TI, namely, a GF-model-like band and $\lambda < 0$.

Finally, we comment on the possibility of lattice distortion in the pyrochlore oxides. For example, though in $Sn_2Nb_2O_7$ there is no experimental signature of lattice distortion from the cubic phase, $Pb_2Nb_2O_7$ suffers a lattice distortion in low temperature [30]. Lattice distortion can greatly affect the topological properties. Zhang et al. have shown that $Tl_2Nb_2O_7$ can transform from a semi-metal to a TI by in-plane strain [18]. Since the topological properties are not dependent on the detail of the band structure, it is highly possible that the TI is also induced by in-plane strain in $Tl_2Ta_2O_7$, $In_2Nb_2O_7$ and $In_2Ta_2O_7$, which have similar band structures.

## 4. Conclusions

We calculated the band structure of six pyrochlore oxides, $A_2B_2O_7$ (A = Sn, Pb, Tl; B = Nb, Ta). They have quasi-flat valence bands that are characteristic of the Guo–Franz flat-band model. Spin–orbit interaction partially lifts the degeneracy at the $\Gamma$ point. This results in a strong topological insulator in the s1 compound if the split energy levels satisfy $E(\Gamma_7) < E(\Gamma_8)$. We found only $Sn_2Nb_2O_7$ satisfies $E(\Gamma_7) < E(\Gamma_8)$ in the above six compounds, though this is an s2 compound. We propose that the virtual compound $(In,Sn)_2(Nb,Zr)_2O_7$ may have all of the desired properties for a TI, i.e., a Guo–Franz-model-like band, an s1 compound and $E(\Gamma_7) < E(\Gamma_8)$.

**Author Contributions:** Conceptualization: I.H. and T.Y.; methodology: I.H.; software: I.H.; investigation: I.H.; data curation: I.H.; writing—original draft preparation: I.H.; writing—review and editing: T.Y.; visualization: I.H.; project administration: I.H.; funding acquisition: I.H. and T.Y. All authors have read and agreed to the published version of the manuscript.

**Funding:** This research was funded by the Japan Society for the Promotion of Science, grant number 19K03731.

**Acknowledgments:** The authors would like to thank Y. Higashi, M. Suzuki, Y. Yanagi, K. Kawashima and H. Aoki for fruitful discussions.

**Conflicts of Interest:** The authors declare no conflict of interest.

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
