# Peer review of "Possible Three-Dimensional Topological Insulator in Pyrochlore Oxides"

_symmetry, doi:10.3390/sym12071076_

Round 1

Reviewer 1 Report

Authors report the first-principles study on band structures of A2B2O7 pyrochlore oxides and identify strong topological insulator driven by spin-orbit coupling in the submitted manuscript. I think the topic is interesting, and the study is carefully conducted and presented clearly. Therefore, I would like to recommend this manuscript for publication in Symmetry.

Author Response

Thank you for appreciating our work. We are very glad that Reviewer 1 recommended our manuscript for publication in Symmetry.

According to the instructions by other reviewers, we revised the manuscript. Some typos are also corrected.

Reviewer 2 Report

The authors calculate the band spectrum of a series of pyrochlore oxides within the density functional approach with augmented plane waves, aiming to recognize possible cases in which the spin-orbit coupling could generate the gap at the Fermi level and topologically protected three dimensional insulatoring state. Their results suggest that (at least) one compound is a favorable candidate, hole-doped Sn2Nb2O.

The work is well-written. (Some, more or less minor, suggestions towards its better readibility are listed below.) It would be helpful in the targeted search for topological insulators that could be, also due to their general propeties (chemical stability etc), acceptable for further examinations towards applications. 

Having in mind all mentioned reasons, it is my opinion that the work can be recommended for publication in Symmetry. 

Suggestions:

  • Eq. 1; What is the meaning of superscripts in d1,2?
  • Lines 65, 151; A sort description of the meaning of topological indices (0;000) would be helpful for a non-specialized reader; 
  • Line 100; What is the meaning of the parameter u? Also, the definition of Hellmann-Feynman force; 
  • Line 152: Some misprint at the beginning of the sentence "We see that ..."?
  • To facilitate the reader's insight into the crystal lattice, positions of atoms and their orbitals, etc, it would be advisable to include appropriate Figure(s).

Reviewer 3 Report

The authors present the results of numerical band structure calculations for six pyrochlore oxides. They propose a compound containing Sn as a candidate for a strong topological insulator.

The manuscript is well organized and has reasonable discussions of physical systems, but it is written in a very careless way. I will mention just a few examples of these negligences. In Eq.(1) the second term describes spin-orbit coupling. A double bracket sign is used for the next-nearest neighbor sites. Does it mean that nearest-neighbor sites lack spin-orbit coupling? When the authors discuss topological indices below Eq.(2) they present four zeros without any explanation. In the previous sentence, they write that parameter $\lambda <0$ twice?

When they discuss computational methods a parameter $RK_{max}$ is mentioned without any effort to explain it. The words “second-variational manner” also do not bring much information to the reader.

The language of the manuscript has a lot of misprints and simple errors.

To summarize, this manuscript can be published only after text and language revision.
